# Metabolomics Analyses in High-Low Feed Efficient Dairy Cows Reveal Novel Biochemical Mechanisms and Predictive Biomarkers

**DOI:** 10.3390/metabo9070151

**Published:** 2019-07-23

**Authors:** Xiao Wang, Haja N. Kadarmideen

**Affiliations:** Quantitative Genomics, Bioinformatics and Computational Biology Group, Department of Applied Mathematics and Computer Science, Technical University of Denmark, Richard Petersens Plads, Building 324, 2800 Kongens Lyngby, Denmark

**Keywords:** metabolomics, residual feed intake, gene-metabolite network, dairy cattle

## Abstract

Residual feed intake (RFI) is designed to estimate net efficiency of feed use, so low RFI animals are considered for selection to reduce feeding costs. However, metabolic profiling of cows and availability of predictive metabolic biomarkers for RFI are scarce. Therefore, this study aims to generate a better understanding of metabolic mechanisms behind low and high RFI in Jerseys and Holsteins and identify potential predictive metabolic biomarkers. Each metabolite was analyzed to reveal their associations with two RFIs in two breeds by a linear regression model. An integrative analysis of metabolomics and transcriptomics was performed to explore interactions between functionally related metabolites and genes in the created metabolite networks. We found that three main clusters were detected in the heat map and all identified fatty acids (palmitoleic, hexadecanoic, octadecanoic, heptadecanoic, and tetradecanoic acid) were grouped in a cluster. The lower cluster were all from fatty acids, including palmitoleic acid, hexadecanoic acid, octadecanoic acid, heptadecanoic acid, and tetradecanoic acid. The first component of the partial least squares-discriminant analysis (PLS-DA) explained a majority (61.5%) of variations of all metabolites. A good division between two breeds was also observed. Significant differences between low and high RFIs existed in the fatty acid group (*P* < 0.001). Statistical results revealed clearly significant differences between breeds; however, the association of individual metabolites (leucine, ornithine, pentadecanoic acid, and valine) with the RFI status was only marginally significant or not significant due to a lower sample size. The integrated gene-metabolite pathway analysis showed that pathway impact values were higher than those of a single metabolic pathway. Both types of pathway analyses revealed three important pathways, which were aminoacyl-tRNA biosynthesis, alanine, aspartate, and glutamate metabolism, and the citrate cycle (TCA cycle). Finally, one gene (2-hydroxyacyl-CoA lyase 1 (*+HACL1)*) associated with two metabolites (-α-ketoglutarate and succinic acid) were identified in the gene-metabolite interaction network. This study provided novel metabolic pathways and integrated metabolic-gene expression networks in high and low RFI Holstein and Jersey cattle, thereby providing a better understanding of novel biochemical mechanisms underlying variation in feed efficiency.

## 1. Introduction

Gross feed efficiency (GFE), expressed as the ratio of output (e.g., milk yield) to feed input (e.g., total dry matter intake (DMI)), was used to define feed efficiency in dairy cows [1]. The expression of GFE corresponding to the feed conversion ratio is easy to be measured but has some limitations [2]. Residual feed intake (RFI) is designed to estimate the net feed efficiency by comparing actual and predicted feed intake. Koch et al. (1963) [3] proposed RFI to take into account energy costs for body maintenance and weight gains for determining efficiency of feed use in the growing beef cattle. Low RFI animals are considered efficient and eat less, whereas high RFI animals are inefficient and eat more than the average of their production level [4].

Several biological factors are likely to contribute to the variations in RFI, including breed type, variation in genetic makeup, variation in metabolic processes or gene expression, diet composition, and other environmental factors. Waghorn et al. (2006) [5] reported that around 5% unit digestion variations exited between individual cows. Previous studies have shown higher feed efficiency in Jersey cows than in Holstein cows in Denmark [4,6], which are consistent with the feed efficiency studies for dairy cattle from other countries [7]. However, Aikman et al. (2008) [8] found that Jersey cows have similar intake capacity compared with Holstein cows, probably because cows were selected for equal expected milk energy yield. In Danish dairy cows, Shetty et al. (2016) [9] concluded that there are better prediction accuracies by predicting the RFI by excluding the random effects for validation. Li et al. (2017) [10] also found that neglect of the lactation stage could result in a naive assessment of RFI. Salleh et al. [11,12] analyzed liver transcriptomics data to construct gene co-expression networks for low and high RFI groups of cows and detected expression quantitative trait loci (eQTLs are genetic variants that regulate gene expression levels in low/high RFI cows), both studies leading to candidate genes for RFI [11,12].

Metabolomics has been increasingly used to measure the dynamic metabolic responses in dairy cows [13,14]. A metabolic pathway links series of chemical reactions in a cell. The pathways of metabolism enable us to break down or synthesize many important molecules and initiate efficient reactions quickly. Metabolomics and pathways have been characterized for pregnant dairy cows to seek biochemical insight into possible biological modules related to early pregnancy [15]. Blood plasma metabolome can be cheaply measured on a large number of animals compared to other -omics profiles used to predict RFI (e.g., transcriptomics of RNA sequencing, as in Salleh et al. (2017) [16]). Furthermore, metabolites are further downstream in the biological processes from DNA through RNA to proteins, and hence are closer to observable phenotypes. If metabolites are highly predictive of the RFI phenotype, it could be used in animal selection of low RFI for better herd management or for breeding.

Until now, metabolic profiling of cows and availability of predictive metabolic biomarkers for RFI are scarce. Therefore, one of the objectives of this study was to generate a better knowledge of underlying metabolic mechanisms that possibly characterize low and high RFI in Jerseys and Holsteins. The other objective was to identify potentially predictive metabolic biomarkers for RFIs. These objectives were achieved by designing a cattle experiment to provide an evaluation and temporal comparison of the plasmatic metabolome analysis. By combining pathway analysis methods, a gas chromatography-mass spectrometry (GC-MS) system was used in the identification of each metabolite to generate a better understanding of the metabolic mechanisms occurring in Nordic dairy cows. Metabolite set enrichment analysis (MSEA) was used to investigate a set of functionally related metabolites. A novel two-way integration of metabolomics and transcriptomics profiles in low and high RFI cows was used to link their genomic variation with their metabolomics variation and report upregulated and downregulated gene-metabolite combinations.

## 2. Results

### 2.1. Statistics of the Identified Metabolites and their Pearson Correlation Coefficients

Citric acid, heptadecanoic acid, hexadecanoic acid, octadecanoic acid, palmitoleic acid, pentadecanoic acid, and tetradecanoic acid of 37 identified metabolites from 20 plasma samples showed high values in Figure 1. Two breeds (Jersey and Holstein cows) and two RFIs (low and high) were found to show statistically significantly different values (*P* < 0.001) in these seven identified metabolites (Figure 1) after a Chi-squared test. The precision (%) of 4-amino-benzoic acid and malic acid were more than 50%, but their values were nearly 0 (Figure 1).

The significant correlations of Pearson correlation coefficients (PCCs) (*P* < 0.001) between two metabolites were observed in one big group, including asparagine, methionine, glycine, serine, histidine, lysine, ornithine, tryptophan, tyrosine, alanine, proline, and threonine, and two small groups, including pentadecanoic acid, hexadecanoic acid, heptadecanoic acid, and octadecanoic acid in one group and glutamine, phenylalanine, valine, isoleucine, and leucine in another group (Figure 2). We also found that nearly all of the significant correlations were positive and their PCCs were higher than 0.5.

### 2.2. Metabolite Clusters and Comparisons between Low and High RFIs

Figure 3 provides clusters of metabolites with low and high RFI groups (all animals). In general, the concentrated values ranged from −1.5 to +1.5 after scaling by metabolite-wise in columns. Among 34 identified metabolites, for which scaled values higher than the limit of detection (LOD) score, three main clusters were observed in the heat map. The lower cluster were all from fatty acids, including palmitoleic acid, hexadecanoic acid, octadecanoic acid, heptadecanoic acid, and tetradecanoic acid (Figure 3). The values of these metabolites were medium for low RFIs. High RFI in Jersey and Holstein cows showed completely different values. Generally, metabolites of low RFI Holstein cows displayed higher values than the other three groups (high RFI Holstein, low RFI Jersey, and high RFI Jersey) (Figure 3)

The first component (Component 1) and second component (Component 2) of partial least squares-discriminant analysis (PLS-DA) explained 61.5% and 11% variations of all 34 metabolites, respectively (Figure 4a). The high RFI group was shown in the horizontal line, while the low RFI group was in the vertical direction. It was observed that all of the metabolites were relatively more over-represented in the Jersey group than the Holstein group. Additionally, a good division appeared between Jersey (J) and Holstein (H) breeds (Figure 4a). The loading plot results showed that eight metabolites (citric acid, heptadecanoic acid, hexadecanoic acid, octadecanoic acid, palmitoleic acid, pentadecanoic acid, tetradecanoic acid, and valine) caused the separation between different breeds and RFI groups in PLS-DA (Figure 4b). As a supervised method, PLS-DA is more susceptible to overfitting, so it needs to be verified. The permutation results here confirmed that the PLS-DA was valid with a *P* value (0.012) < 0.05 after 1000 permutation tests (Figure 4c). From the box plots of α-ketoglutarate and succinic, the fold change of low RFIs showed relatively higher values than the fold change of high RFIs (Figure 4d,e).

Among 15 metabolites in the amino acid group, five metabolites in the tricarboxylic acid group and six metabolites in the fatty acid group, differences between low and high RFIs were significant (*P* < 0.001) in the fatty acid group (Table 1). If all three groups were considered together, two RFIs also showed significant differences (*P* < 0.001) (Table 1).

Statistical analysis of breed, parity, and RFI for 34 identified metabolites showed that the breed had significant factorial effect on leucine, ornithine, pentadecanoic acid, and valine, while the RFI exited no significant effect (Table 2).

### 2.3. Significant Metabolic Enrichments, Pathways, and Networks

Metabolite sets were significantly enriched (*P* < 0.05) in four organs or tissues (prostate, mitochondria, peroxisome, and skeletal muscle) after MSEA, based on organ, tissue, and subcellular localizations, with 27 metabolites located in the prostate (267 metabolites in total) as the most significant set (Figure 5). The most significant metabolic pathway, using *Bos taurus* as a library, was aminoacyl-tRNA biosynthesis (bta00970), in which 17 metabolites were involved (Table 3). A total of 14 metabolites (alanine, asparagine, aspartic acid, glycine, isoleucine, leucine, lysine, methionine, phenylalanine, proline, serine, threonine, tyrosine, and valine) from amino acids and three other metabolites (histidine, glutamine, and tryptophan) comprised the aminoacyl-tRNA biosynthesis pathway in our study.

The integrated gene-metabolite pathway analysis showed that three pathway impact values increased higher than the single metabolic pathway analysis, while test powers and matched metabolite ratios decreased lower than before (Figure 6). We found three important gene-metabolite pathways with most powerful testing values and the largest pathway impact values using *Homo sapiens* as the library, which were aminoacyl-tRNA biosynthesis (18/87), the alanine, aspartate and glutamate metabolism (9/56), and the citrate cycle (TCA cycle) (7/50) (Figure 6b). A total of 24 metabolites were detected in these three pathways, including asparagine, aspartic acid, cis-aconitic acid, citric acid, fumaric acid, glutamic acid, glutamine, glycine, histidine, isocitric acid, isoleucine, methionine, leucine, lysine, oxoglutaric acid, phenylalanine, proline, pyruvic acid, serine, succinic acid, threonine, tryptophan, tyrosine, and valine. Moreover, eight metabolites of them were doubly detected among the three pathways (asparagine, aspartic acid, fumaric acid, glutamic acid, glutamine, oxoglutaric acid, pyruvic acid, and succinic acid). In addition, the gene-metabolite interaction network revealed only one network that was a one upregulated gene (*+HACL1*) associated with two downregulated metabolites (α-ketoglutarate and succinic acid) (Figure 7a). The two metabolites involved in the network showed higher values of α-ketoglutarate than succinic acid (Figure 7b).

## 3. Discussion

This metabolomics study on dairy cattle feed efficiency was designed to provide key biological insights into differences between a number of factors, such as breed, parity, etc., but most importantly an association between feed efficiency and key identified metabolites in the blood plasma of dairy cows. Given that feed efficiency is a critically important production trait in the breeding goals of both dairy and beef cattle breeding [16,17,18], a cost efficient and easy way to measure predictive metabolite biomarkers for feed efficiency would be helpful. We designed this metabolomics study for an evaluation and temporal comparison of the plasmatic metabolome analysis by low or high RFI animals. In this study, we restricted the analyses by only using a high-concentrated diet, as the dataset is too small to study all the interactions.

### 3.1. Plasma Metabolites of Nordic Dairy Cattle

A combination of the features of gas chromatography and mass spectrometry gives GC-MS coupling a high throughput, robustness, and unmatched comprehensiveness for different small molecule classes [19,20]. Zhou et al. (2016) [21] has determined 40 metabolic biomarkers, including 18 amino acids by isotope dilution coupled with the GC-MS method in cow plasma samples for circulating amino acids with a different liver functionality index. In this study, we achieved 37 identified metabolites, including three identified metabolites with values lower than the LOD from approximately 200 spectra peaks using the GC-MS system (Figure 1). They are all linked to 255 bovine plasma-associated metabolites, according to the online livestock metabolome database (LMDB) (http://www.lmdb.ca). Annotated metabolites were primarily matched with the acquired MS spectra based on the library, thus, the annotation might be incorrect, but likely a similar structure. The Chemical Analysis Working Group of the Metabolomics Standards Initiative (MSI) [22] developed the definitions of metabolite annotation and identification. MSI identification requires exhaustive analytical validation to be the most rigorous, whereas annotation does not need it in the categorical scoring system as metabolite annotation is the tentative metabolite candidate to the signal [23]. Therefore, annotated metabolites were not used in this study. Afterwards, the identified metabolites tend to be a small size after rigorous analytical validation to exclude the annotated metabolites.

The identified metabolites of plasma from our study were mainly located in prostate, mitochondria, and peroxisome (Figure 5). Fransen et al. (2017) [24] reviewed that the concerted action of peroxisomes and mitochondria is associated with the diverse cellular metabolic and signaling processes. In addition, prostate, mitochondria, and peroxisome metabolically interact with each other, and mitochondria and peroxisomes play major roles in cell metabolism as ubiquitous organelles, especially in terms of a fatty acid metabolism, which might be related to feed utilization [25].

### 3.2. Key Metabolic Pathways after Single and Integrated Analysis

The previous study has already evaluated metabolomics for the prediction of production traits in cattle, such as RFI [26]. Metabolomics of RFI study in beef cattle reported the plasma metabolites in significant association with RFI, which could be used to predict RFI with high accuracy [27]. Our results also revealed a good division of metabolites between Jersey and Holstein cows combining with low and high RFIs (Figure 4). A total of 34 identified metabolites in four groups of low and high RFI Jersey and Holstein cows were clustered in three main categories, especially for the group of low RFI Holstein cows showing relatively higher values (Figure 3). Karisa’s study found a significant pathway of the citrate cycle (TCA cycle) [27], which is consistent with our results from both single metabolic and gene-metabolite pathway analysis (Figure 6). Pyruvic acid is yielded by glycolysis when sugar breakdown generates the acetyl-CoA as the starting point of the TCA cycle. Other metabolites (cis-aconitic acid, fumaric acid, isocitric acid, oxoglutaric acid, citric acid, and succinic acid) then enter the TCA cycle in a step and finally form a citrate [28]. In addition, cis-aconitic acid and citric acid of the TCA cycle were in strong correlations in our study (Figure 2).

Aminoacyl-tRNA biosynthesis is characterized by the cognation of transfer RNA (tRNA) bonding to its amino acid chemically during protein synthesis. The previous result has indicated that ArgRS, as the unique form of aminoacyl-tRNA synthetases, was essential for normal growth and protein synthesis in mammalian cells [29]. Aminoacyl-tRNA synthetase can cause protein mistranslation and affect cellular physiology and development [30]. Additionally, gene mutations of aminoacyl-tRNA synthetase resulted in neuropathies and myopathies of human beings [31,32].

### 3.3. Metabolic Networks for Gene Expressions and Metabolites

A total of two metabolites that were included in the metabolic network are expected to have an effect on feed efficiency (Figure 7). 2-hydroxyacyl-CoA lyase 1 (+*HACL1*), the first peroxisomal enzyme in mammals dependent on thiamin pyrophosphate (TPP), was negatively associated with α-ketoglutarate and succinic acid in our results. α-ketoglutarate can determine the overall rate of the TCA cycle and modulate protein synthesis and bone development as an important source of amino acids for collagen synthesis in the cell and organism [33]. Succinic acid is also an intermediate compound of the TCA cycle, which links and regulates energy metabolism like an adenosine triphosphate (ATP) formation [34]. Through the mitochondrial Gamma-aminobutyric acid (GABA) transaminase, GABA forms glutamate and succinate semialdehyde with transamination of α-ketoglutarate [34] while it is converted from glutamate by the cytosolic glutamate decarboxylase [35]. Furthermore, a genome-wide association study (GWAS) revealed that RFI was in association with significant single nucleotide polymorphisms (SNPs) of the *GABRR2* gene, which can encode a receptor of GABA [36]. The network of the gene (+*HACL1*) and metabolites (α-ketoglutarate and succinic) involved in the key metabolic pathway of the TCA cycle (Figure 6) might be the potential biochemical mechanisms responsible for feed efficiency of dairy cows.

### 3.4. Implications

To understand the complexity of metabolomics information, network construction based on systems biology/systems genomics should be performed with other multi-omics data types, as described in Kadarmideen (2014) [37] and Suravajhala et al. (2016) [38]. The networks of our study provide a good connection between metabolite levels and gene expression levels. Goldansaz et al. (2017) [26] proposed to store and categorize livestock metabolome information into a standardized format for future livestock research, as they found that fewer than 30 candidate biomarkers were observed in most reports. Most metabolomic results only provided relative metabolite trends, but several cattle studies reported useful or verifiable biomarker data [26]. However, our study provided the potentially predictive metabolic biomarkers for RFI using both linear regression models and the integrated network analyses that combined metabolite data with gene expression data, showing consistency with metabolic regulations in feed efficiency. Overall, the majority of livestock studies used relatively small sample sizes, thus were hard to achieve the good quality biomarkers, especially compared to the human biomarker standards [39].

## 4. Materials and Methods

### 4.1. Animals and Data

A total of 10 Jersey and 10 Holstein cows from a Danish cattle research center (DCRC) were used in this study. The details of animals used in this feeding experiment according to the breed, parity, actual RFI value, RFI group (low or high), and the allocation of the diet concentrations (CDs) are described in our own previous study [16]. This previous study related whole genome-wide gene expression (transcriptomics) profiles with RFI, and the current study is focused on metabolites in plasma and its association with different parameters. All phenotypic data, classification of animals, experimental parameters, and transcriptomic data are already publicly made available in our own previous study [16], except the metabolomics data. Selections of 10 Jersey and 10 Holstein cows as low-RFI and high-RFI groups are given in Table 4, and all cows were only fed on high CDs. The actual RFI value was defined by a one-step approach using the method from Tempelman’s study [40], as reported by Salleh et al. (2017) [16]. A total of two cows, including one low RFI cow and one high RFI cow, were raised in one block, so the cow identity cards (IDs) were paired (Table 4). The cow was defined as low RFI when its actual RFI value was larger than the other cow in the same block, and then the paired cow was defined as high RFI, accordingly.

### 4.2. Metabolomics for Plasma

Blood samples were collected to heparin tubes from the jugular vein of each cow at approximately 7:00 am. The heparin tubes were then centrifuged at 4999 g and 4 °C for 7 min to separate the plasma immediately. Afterwards, plasma aliquot was stored at −20 °C until further metabolic analysis. As expected, the metabolites in blood are connected with the metabolites in the gastrointestinal tract to complete the biochemical reactions based on the blood circulation system. In addition, Wikoff et al. (2009) revealed a significant effect of gut microflora on mammalian blood metabolites [41]. In this study, the gas chromatography-mass spectrometry (GC-MS) system was applied for identifying metabolites. GC-MS systems can only analyze volatile compounds, so chemical derivatization of nonvolatile compounds is required. Using methyl chloroformate (MCF) without any prior treatment, all plasma samples were derivatized to convert amino and nonamino organic acids into volatile carbamates and esters before GC-MS analysis. Accordingly, most metabolites of the central carbon metabolism were presented as the key intermediate of the cell metabolism, even if this treatment is limited to compounds presenting amino and/or carboxyl groups. We took a small aliquot from each sample to a mixed pool, and the pooled sample was tested for matrix effects. After quality control (QC) without the matrix effect, each sample was performed using GC-MS analysis. More compounds and cleaner MS spectra were extracted using the PARAllel FACtor analysis 2 (PARAFAC2) model [42] by MS-Omics software. Due to the derivatization, some extra peaks of compounds originated from the impurities in solvents, thus, these redundant peaks were removed. The GC-MS experiment was completed in the MS-Omics company (Copenhagen, Denmark) using their standard system parameters.

In the GC-MS system, descriptive power (DP) was calculated as the ratio of the standard deviation between the experimental samples and QC samples. Generally, variables with a DP ratio higher than 2.5 are most likely to describe variations related to the experimental design. The percentage of precision (%) was also calculated to denote the relative standard deviation between QC samples, as a measure of the certainty of the individual compounds. The limit of detection (LOD) was listed to indicate the lowest value of a compound that the method enabled to detect.

Approximately 1.5-gigabyte raw data of metabolites from 20 plasma samples were generated and extracted with approximately 200 spectra peaks. Finally, 84 compounds were identified or annotated at three different levels, which were 37 identified metabolites, 34 annotated metabolites, and 13 unknown compounds. The identified metabolites were defined by comparing authentic chemical standards to the retention time and mass spectra. However, the annotated metabolites were primarily based on a library matching the acquired MS spectra with the National Institute of Standards and Technology (NIST) library. The 37 identified metabolites contained 15 amino acids (alanine, asparagine, aspartic acid, glycine, isoleucine, leucine, lysine, methionine, ornithine, phenylalanine, proline, serine, threonine, tyrosine, and valine), 5 tricarboxylic acids (α-ketoglutarate, citric acid, lactic acid, pyruvic acid, and succinic acid), 6 fatty acids (heptadecanoic acid, hexadecanoic acid, octadecanoic acid, palmitoleic acid, pentadecanoic acid, and tetradecanoic acid), and 11 other metabolites. Most of the values of 37 identified metabolites were higher than the values of LOD except 4-amino-benzoic acid, cystine, and, phosphoenolpyruvate, which were all less than LOD, thus, this study only used 34 identified metabolites for the further analysis.

### 4.3. Statistical Analysis

The differences between low and high RFIs were compared with the Student’s t-test among amino acid, tricarboxylic acid, and fatty acid groups. The Pearson correlation coefficient (PCC) of 34 identified metabolites was calculated and visualized by R package corrplot (version 0.84), and each pair of input metabolites were tested to calculate the *P* values. These metabolites were also hierarchically clustered by Ward’s method [43] in a heat map, according to the Euclidean distance measure. The values were scaled metabolite-wise in columns for heat map visualization. Partial least squares-discriminant analysis (PLS-DA), a heat map for averaged metabolite clustering in four groups (i.e., Jersey-Low, Jersey-High, Holstein-Low, and Holstein-High), and a log of fold change (logFC) between low and high RFI groups were calculated by a web-based tool, MetaboAnalyst [44]. A total of 34 identified metabolites were performed in the linear regression model:y=μ+breed+parity+RFI+e,
where y is the value of the metabolite, μ is the intercept, breed is the Jersey and Holstein cows, parity is from 1 to 3, RFI is the actual RFI value as the covariate and e is a residual.

### 4.4. Metabolite Enrichment and Pathway Characterization

Metabolite set enrichment analysis (MSEA) was used to investigate a set of functionally-related metabolites in this study. Location-based metabolite sets were selected as a metabolite library for MSEA. Over-representation analysis (ORA) was implemented to evaluate whether a particular metabolite set is over-represented using the hypergeometric test. In the metabolic pathway analysis, we used *Bos taurus* as the library and Fishers’ exact test for ORA. Relative betweenness centrality was selected for the node importance measure and the pathway impact value calculation in the topological analysis. The pathway impact in each pathway was calculated as the sum of importance measures of the matched metabolites divided by the sum of the importance measures of all the metabolites [45]. MSEA and pathway analysis were performed by MetaboAnalyst [44].

### 4.5. Integration of Metabolomics and Transcriptomics Profiles in Low and High RFI Groups

We also used the MetaboAnalyst tool for the integrative analysis of metabolomics and transcriptomics to create networks [46]. The gene-metabolite interaction network aims to explore interactions between functionally-related genes and metabolites. Due to the species limitation of the MetaboAnalyst tool, we used *Homo sapiens* as the library for the integrated analysis. logFC of genes and metabolites between low and high RFIs were used for the regulating direction in network analysis. In our study, an upregulated and downregulated gene/metabolite was defined when logFC was positive and negative, respectively, after the low and high RFI comparisons. The transcriptomics data was downloaded from publicly available information in our earlier study [16]. Salleh et al. (2017) [16] revealed potential regulatory genes (adjusted *P* value < 0.05) for feed efficiency after comparisons between low and high RFIs in Nordic dairy cattle.

## 5. Conclusions

Feed efficiency, as measured by RFI, is important for profitability and sustainability of dairy cattle production. If metabolites are highly predictive of the RFI phenotype, it could be used in a selection of animals with low RFI for better herd management or for breeding. In this study, we found differences between low and high RFI animals in the fatty acid group (*P* < 0.001). Among 34 identified metabolites, there is clearly a significant difference between breeds. As expected, due to differences in their RFI; however, the association of individual metabolites (leucine, ornithine, pentadecanoic acid, and valine) with the RFI status were only marginally significant or not significant due to lower sample size. This study also provided better understanding of novel biochemical mechanisms underlying variations in feed efficiency within and between two dairy breeds. Key potential metabolic biomarkers (aminoacyl-tRNA biosynthesis, alanine, aspartate and glutamate metabolism, and the citrate cycle (TCA cycle)) were reported here that could be further validated in larger populations and used as a pre-screening tool for the selection of animals with better feed efficiency. Finally, as genomic selection methods improve to integrate predictors other than genomic data, we see a potential for reported metabolic predictors here to contribute to this development.

## Figures and Tables

**Figure 1 metabolites-09-00151-f001:**
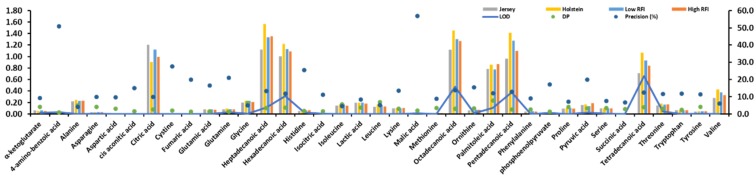
Statistical description of 37 identified metabolites. Limit of detection (LOD). Descriptive power (DP) that is using the y-axis with precision (%).

**Figure 2 metabolites-09-00151-f002:**
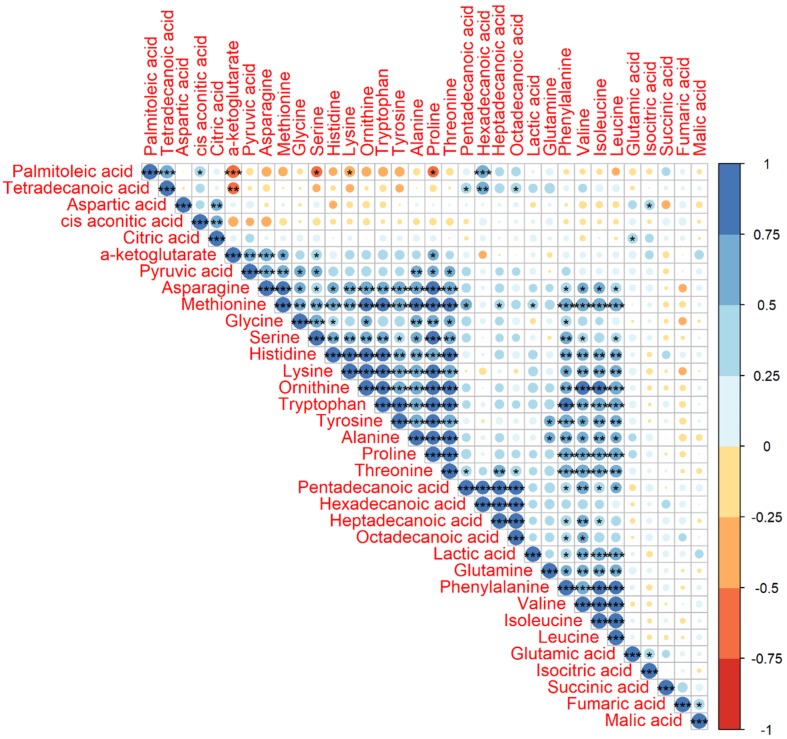
Pearson correlation coefficient (PCC) analysis of 34 identified metabolites. Note: ^*^ indicates *P* value < 0.05, ^**^ indicates *P* value < 0.01, and ^***^ indicates *P* value < 0.001. The number on the right bar indicates the PCCs from −1 to 1. The PCCs of the diagonal were 1.

**Figure 3 metabolites-09-00151-f003:**
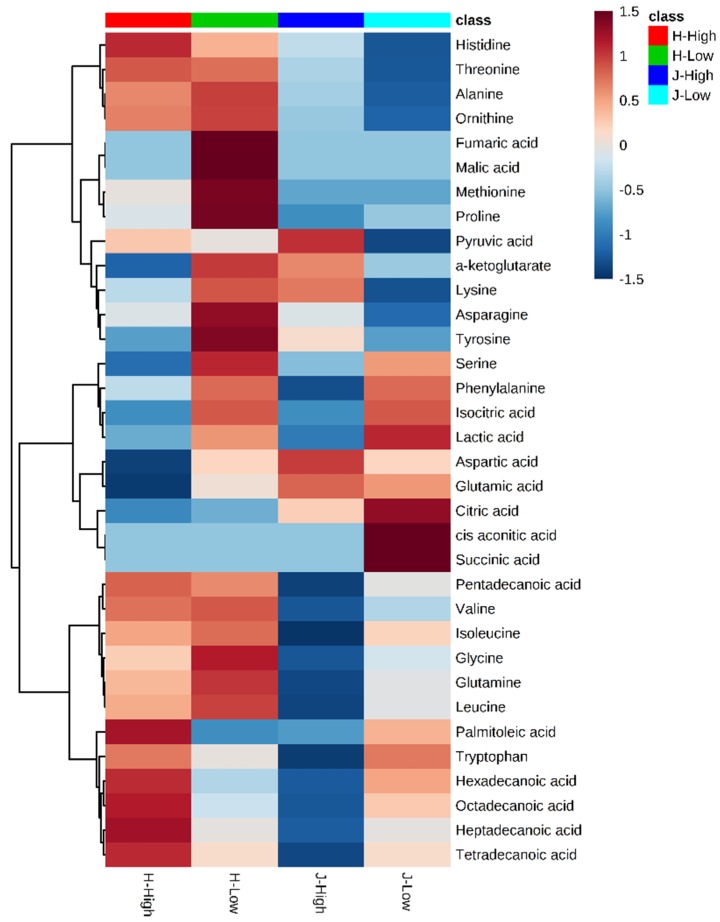
Heat map for hierarchical clustering of 34 identified metabolites between low and high residual feed intakes (RFIs). Note: J/H with numbers indicates Jersey/Holstein ID. Low and High indicate low and high RFIs.

**Figure 4 metabolites-09-00151-f004:**
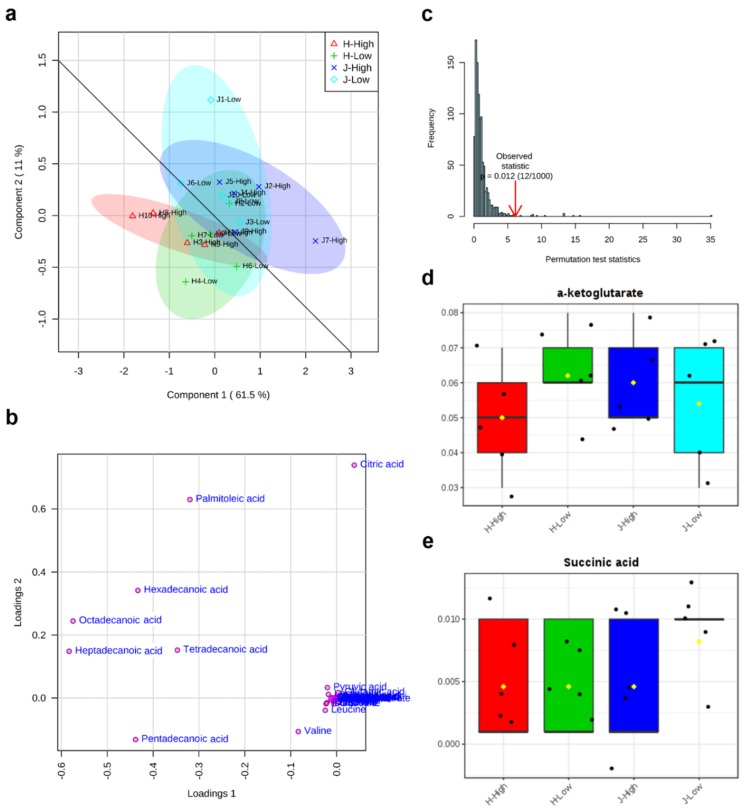
Partial least squares-discriminant analysis (PLS-DA) of 34 identified metabolites between low and high residual feed intakes (RFIs) of Jersey and Holstein cows. (**a**) 2 dimensional score plot of PLS-DA; (**b**) loading plot of metabolite separations in PLS-DA; (**c**) permutation test (n = 1000) for PLS-DA; (**d**) and (**e**) box plots of α-ketoglutarate and succinic acid for relative fold change after PLS-DA. Note: J/H with number indicates Jersey/Holstein ID. Low and High indicate low and high RFIs. The overlapped metabolites in the cluster of (**b**) include α-ketoglutarate, alanine, asparagine, aspartic acid, cis aconitic acid, fumaric acid, glutamic acid, glutamine, glycine, histidine, isocitric acid, isoleucine, lactic acid, leucine, lysine, malic acid, methionine, ornithine, phenylalanine, proline, pyruvic acid, serine, succinic acid, threonine, tryptophan, tyrosine.

**Figure 5 metabolites-09-00151-f005:**
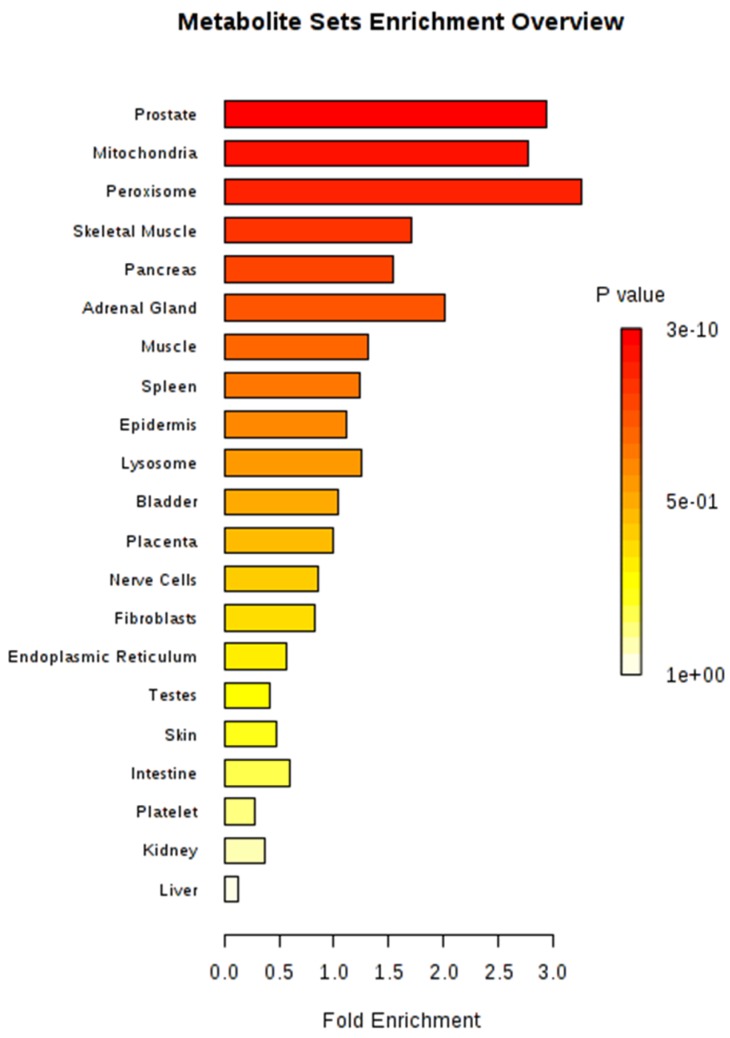
Metabolite set enrichment analysis (MSEA) for 34 identified metabolites.

**Figure 6 metabolites-09-00151-f006:**
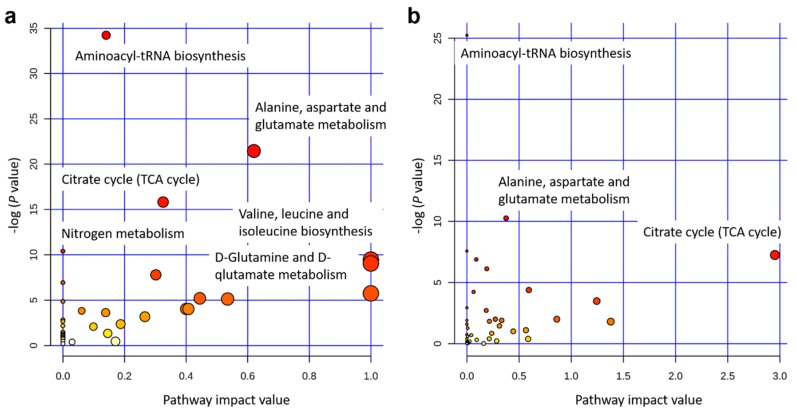
Pathway analysis for 34 identified metabolites using *Homo sapiens* as the library. (**a**) Single metabolic pathway analysis. (**b**) Integrated metabolic pathway analysis from combined metabolome and transcriptome studies. Note: The color and size of the circles indicate the log (*P* value) and the matched metabolite ratio, respectively, for each pathway. The pathway impact value is calculated as the sum of importance measures of the matched metabolites divided by the sum of the importance measures of all metabolites.

**Figure 7 metabolites-09-00151-f007:**
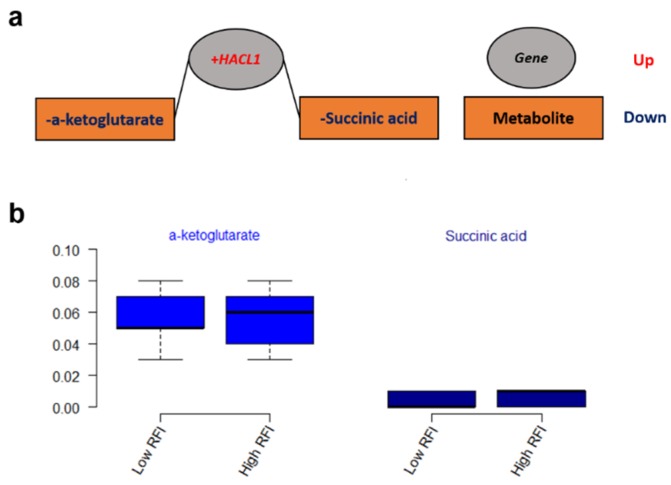
(**a**) The gene-metabolite interaction network from the combined metabolome and transcriptome studies. (**b**) The values of metabolites that are involved in the network.

**Table 1 metabolites-09-00151-t001:** The differences between low and high residual feed intakes (RFIs) among amino acids, tricarboxylic acids, fatty acids, and all of them.

RFI	Amino Acid (Mean ± SE)	Tricarboxylic Acid (Mean ± SE)	Fatty Acid (Mean ± SE)	All 26 Metabolite (Mean ± SE)
Low	0.12 ± 0.03	0.29 ± 0.12	1.10 ± 0.12^***^	0.38 ± 0.15^***^
High	0.11 ± 0.03	0.31 ± 0.14	1.52 ± 0.23^***^	0.48 ± 0.22^***^

Note: Standard error (SE). ^***^ indicates *P* value < 0.001.

**Table 2 metabolites-09-00151-t002:** Significant metabolites associated with breed and residual feed intake (RFI) effects.

Metabolite (Mean ± SE)	Leucine	Ornithine	Pentadecanoic Acid	Valine
Breed	−0.05 ± 0.02^*^	−0.02 ± 0.01^*^	−0.51 ± 0.15^**^	0.33 ± 0.05^***^
RFI	0.02 ± 0.01 (*P* = 0.06)	−0.001 ± 0.01 (*P* = 0.7)	0.16 ± 0.08 (*P* = 0.07)	0.04 ± 0.02 (*P* = 0.09)

Note: Standard error (SE). ^*^ indicates *P* value < 0.05, ^**^ indicates *P* value < 0.01, and ^***^ indicates *P* value < 0.001.

**Table 3 metabolites-09-00151-t003:** Significant metabolic pathways (*P* < 0.05) after single pathway analysis using *Bos taurus* as the library.

Pathway Name	Match Status	*P* Value	-log (*P* Value)	FDR	Impact
Aminoacyl-tRNA biosynthesis	17/64	1.4 × 10^−15^	34	1.1 × 10^−13^	0.14
Alanine, aspartate, and glutamate metabolism	9/23	4.8 × 10^−10^	21	2.0 × 10^−8^	0.62
Citrate cycle (TCA cycle)	7/20	1.4 × 10^−7^	16	3.7 × 10^−6^	0.33
Nitrogen metabolism	4/9	3.1 × 10^−5^	10	6.2 × 10^−4^	0
Valine, leucine, and Isoleucine biosynthesis	4/11	7.7 × 10^−5^	9.5	0.0013	1.0
D-Glutamine and D-glutamate metabolism	3/5	1.2 × 10^−4^	9.1	0.0016	1.0
Arginine and proline metabolism	6/44	4.1 × 10^−4^	7.8	0.0048	0.30
Butanoate metabolism	4/20	9.8 × 10^−4^	6.9	0.0099	0
Phenylalanine, tyrosine, and tryptophan biosynthesis	2/4	0.0032	5.8	0.029	1.0
Glyoxylate and dicarboxylate metabolism	3/16	0.0055	5.2	0.044	0.44
Glycine, serine, and threonine metabolism	4/32	0.0059	5.1	0.044	0.53
Cyanoamino acid metabolism	2/6	0.0077	4.9	0.052	0
Methane metabolism	2/9	0.018	4.0	0.10	0.4
Phenylalanine metabolism	2/9	0.018	4.0	0.10	0.41
Glutathione metabolism	3/26	0.021	3.8	0.12	0.061
Cysteine and methionine metabolism	3/28	0.027	3.6	0.13	0.14
Histidine metabolism	2/14	0.042	3.2	0.20	0.27

False discovery rate (FDR).

**Table 4 metabolites-09-00151-t004:** Low/high residual feed intake (RFI) of Jersey and Holstein cows, as described in Salleh et al. (2017) [16].

Breed	RFI	Actual RFI Value	Cow ID	Parity	Breed	RFI	Actual RFI Value	Cow ID	Parity
Jersey	Low	0.80	J1-Low	1	Holstein	Low	−0.03	H2-Low	1
		2.23	J3-Low	3			0.10	H4-Low	2
		0.94	J6-Low	3			0.70	H6-Low	3
		0.46	J8-Low	2			0.89	H7-Low	2
		0.49	J10-Low	1			0.41	H9-Low	1
	High	−0.40	J2-High	1		High	−1.10	H1-High	1
		−0.04	J4-High	3			0.05	H3-High	3
		−1.05	J5-High	3			−0.62	H5-High	3
		−1.71	J7-High	2			−1.05	H8-High	2
		−0.51	J9-High	1			−0.40	H10-High	1

## Data Availability

The datasets used in the current study are available from the corresponding author. H.N.K.: hajak@dtu.dk.

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
