# Peer review of "Metabolomics Analyses in High-Low Feed Efficient Dairy Cows Reveal Novel Biochemical Mechanisms and Predictive Biomarkers"

_metabolites, 2019, doi:10.3390/metabo9070151_

Round 1
Reviewer 1 Report
see the attachment.

Author Response
Reviewer 1
Reviewer 1 comment # 1:
A clarification of biochemical bases, which can contribute to a better attitude of cows is a
meaningful concern also for the improvement of the animal welfare.
That's why I find research in this area interesting.
Research on biochemical mechanisms of different diets is increasing in the literature, but studies on
predictive biomarkers are still rare. The study of metabolomic markers provides insights into the
phenotype of the cow. This is also associated with the feed conversion. The distribution of certain
fatty acids in the blood has not yet been brought to the recovery of high-low-feed efficient dairy
cows. By including further evaluation options such as pathway analysis, the mere determination of
metabolites first gets its complex biochemical background. As a result, the event is considered
more comprehensively and provides space for creating research approaches. Therefore, I consider
the data collected to be worthy of publication.
The paper is written intelligibly and the data presented comprehensible. The conclusion, however,
also presents the findings as well as the untapped potential of future integrative research in this
field.
I recommend the acceptance of the article on the condition of eliminating the small defects =
"minor" as acceptance level.
Author response # 1: Thank you for the comments. We have re-edited the defective parts in the manuscript and improved the written English. Please check the revised manuscript in the attachment.

Reviewer 2 Report
The manuscript “Metabolomics analyses in high-low feed efficient dairy cows reveal novel biochemical mechanisms and predictive biomarkers” is an interesting manuscript. This manuscript identified some important metabolites which could be used as the biomarkers for RFI - an important trait for dairy cattle. I found the manuscript has a good merit for publication. I have some suggestions which the authors might consider for improvement of manuscript.
· The authors should introduce some important aspects of RFI in Denmark in the introduction part, does the country currently use for genomic selection
· Could authors give some details about time of measure of blood metabolites? How the nutrition can influence to the metabolites concentration? Do authors have any comments about the metabolites in blood with metabolites in the gastrointestinal tract? It might be the blood metabolites don’t reflect the actual metabolites for efficient animals.
· The authors also identified 13 unknown compounds, is anyways to make use of them?
· Did authors adjust for multiple testing when running the linear mixed model?
· Some metabolites have very low concentration and might costly to identify, how authors can use them as the biomarkers?
· In the integration with the transcriptomic results, did the methods to generate results similar (for instance the model used and the fixed effects)?
· The authors also need to pay an attention to the spelling and typo-errors.
· Did the authors check the hub (metabolites) or genes in the created network(s)?
Minors
Abstract
· Line 16: might replace “selection purposes” to “selection reducing feed intake” to make it clearly
· Line 17: this study is … should be this study was… to make it consistent to the whole manuscript
· Line 19-20: the author might specify more, how metabolites were analyses and which model used in very short, since it is the core section of the manuscript.
· Line 20: either use a linear regression model or linear regressions models
· Line 24-25: The first and the second principle component… can be written as “two major principle components”…
· Why no results reported for the linear mixed model here
Introduction
Line 42: should remove the recently, because the RFI was termed long ago, calculating can be replaced by subtracting or comparing
Line 47: “biological processes” should be replaced by “factors” since breed type… are not the processes
Line 49: Waghorn et al. (2006) [5] has reported that, remove has
Line 60: “Until now, metabolomics profiles associated with low and high RFI cows are scarce.” Could the authors explore more about it?
Line 61: this study is should written as this study was to make is consistent to the rest of manuscript
Results
The authors might provide the raw concentration results of the metabolite measure, since the figure make is quite difficult to examine the differences in concentrations among them
Line 88: which types of correlation
Line 97 and 102: be consistent in the tense used in the manuscript “Figure 3 provides” and “Figure 3 also revealed”
Line 111: From the figure 4, it is impossible to see the separation between two breeds
Figure 4: The authors should add the name on metabolites in the legend
Line 115-116: please paraphrase
Line 117: why 26
Table 2: please rename table, since it is confused that the RFI also significantly influenced on the metabolite concentrations
Line 146; check for double space
Line 154: should be was
Line 153-155: where we can find the result with HACL1
Discussion
161-174: Might move to introduction to give more space for deeper discussion
Line 178: Are differences between the metabolomics markers and the amino acid concentrations, since I found the authors mixed use of them in the manuscript?
Line 180: 34 or 37?
Could the authors comment about why fewer metabolites were identified in the study compared to the database?
Line 191: reviews to reviewed
Line 193: the prostate is mostly for the male reproduction, how it can link to the RFI in dairy cows or female cows here?
Can the authors have any comments why no metabolite significantly associated with RFI in the liner model test? Any potential biasness could affect the results of the test? How the relationship between the animals?
Line 196: there are many other metabolic studies, the authors might added some other to make the discussion deeper
Line 220: Many studies have linked the Gamma-aminobutyric acid (GABA) transaminase to the RFI as well as the feed intake, the authors might add more references and extend the discussion about them.
How the concentration of the two metabolites in plasma?
Methods
Could the authors explain why the cow number 3 are in the high RFI why the cow number 2 in the low RFI? When computing the RFI, did authors include other effects? It might be useful to give a very short summary of how the RFI was computed and explain in brief how they were was classified as low and high
When the plasma samples were collected? Do authors have any comments of about metabolite concentrations in different time of the day?
Line 268: “37 identified metabolites, 34 annotated metabolites and 13 unknown,” should it be 47
Line 287: were
Line 295: what is the metabolic value?
If the breeds are different in RFI, did the author try to run the model within each breed, since it might be useful to have different set of metabolites for each breed?
Why did not the authors include the unknown metabolites for the analyses, they might be useful in the future when more metabolites were identified?
Conclusions
The conclusion is too long, the authors already had the implication part, I don’t see the need for this part. Otherwise, please shorten it.
Author Response
Reviewer 2
Reviewer 2 comment # 1:
The manuscript “Metabolomics analyses in high-low feed efficient dairy cows reveal novel biochemical mechanisms and predictive biomarkers” is an interesting manuscript. This manuscript identified some important metabolites which could be used as the biomarkers for RFI - an important trait for dairy cattle. I found the manuscript has a good merit for publication. I have some suggestions which the authors might consider for improvement of manuscript.
· The authors should introduce some important aspects of RFI in Denmark in the introduction part, does the country currently use for genomic selection
Author response # 1: Thank you for the comments. We have added more information about RFI in Denmark in the Introduction part. Until now, the large-scale genomic selection for RFI in Denmark is scarce because the recording of phenotype RFI requires equipment that digitally measure real-time feed intake by cows as and when they eat and also record frequency, duration etc. Therefore, it is limited to small number of experimental animals in research experimental stations, like the one we have used in our study. However, it might be realized in the coming years, as the breeding companies are working on large-scale phenotyping and using a larger population size.
In Danish dairy cows, Shetty et al. (2016) [9] concluded that there are better prediction accuracies by predicting the RFI by excluding the random effects for validation. Li et al. (2017) [10] also found that neglect of lactation stage could result in naive assessment of RFI. Salleh et al. [11,12] analyzed liver transcriptomics data to construct gene co-expression networks for low and high RFI groups of cows and detected expression quantitative trait loci - eQTLs (genetic variants that regulate gene expression levels in low / high RFI cows), both studies leading to candidate genes for RFI [11,12].
9. Shetty, N.; Løvendahl, P.; Lund, M. S.; Buitenhuis, A. J. Prediction and validation of residual feed intake and dry matter intake in Danish lactating dairy cows using mid-infrared spectroscopy of milk. J. Dairy Sci. 2016, 100, 253-264.
10. Li, B.; Berglund, B.; Fikse, W. F.; Lassen, J.; Lidauer, M. H.; Mäntysaari, P.; Løvendahl, P. (2017). Neglect of lactation stage leads to naive assessment of residual feed intake in dairy cattle. . J. Dairy Sci. 2017, 100, 9076-9084.
11. Salleh, S. M.; Mazzoni, G.; Løvendahl, P.; Kadarmideen, H. N. Gene co-expression networks from RNA sequencing of dairy cattle identifies genes and pathways affecting feed efficiency. BMC Bioinformatics. 2018, 19, 513.
12. Salleh, S. M.; Mazzoni, G.; Nielsen, M. O.; Løvendahl, P.; Kadarmideen, H. N. Identification of Expression QTLs Targeting Candidate Genes for Residual Feed Intake in Dairy Cattle Using Systems Genomics. J. Genet. Genome Res. 2018, 5, 035.
Reviewer 2 comment # 2:
· Could authors give some details about time of measure of blood metabolites? How the nutrition can influence to the metabolites concentration? Do authors have any comments about the metabolites in blood with metabolites in the gastrointestinal tract? It might be the blood metabolites don’t reflect the actual metabolites for efficient animals.
Author response # 2: Yes, we have added details of metabolite measure in the Materials and Methods part. The nutrition in the diet could influence the metabolite concentrations, so we only use the high concentrated diets (CDs) to feed all animals in this study.
Obviously, the metabolites in blood are connected with the metabolites in the gastrointestinal tract, as they collaborate to complete the biochemical reactions such as Citrate cycle (TCA cycle) based on blood circulation system. In addition, Wikoff et al. (2009) demonstrated a significant effect of gut microflora on mammalian blood metabolites.
Blood samples were collected to heparin tubes from the jugular vein of each cow at approximately 7 a.m. The heparin tubes were then centrifuged at 4999 g and 4°C for 7 minutes to separate the plasma immediately. Afterwards, plasma aliquot was stored at -20°C until further metabolic analysis. As expected, the metabolites in blood are connected with the metabolites in the gastrointestinal tract to complete the biochemical reactions based on blood circulation system. In addition, Wikoff et al. (2009) revealed a significant effect of gut microflora on mammalian blood metabolites [41]. In this study, Gas chromatography - Mass spectrometry (GC-MS) system was applied for identifying metabolites. GC-MS systems can only analyze volatile compounds, so chemical derivatization of nonvolatile compounds is required. With using methyl chloroformate (MCF) without any prior treatment, all plasma samples were derivatized to convert amino and nonamino organic acids into volatile carbamates and esters before GC - MS analysis. Accordingly, most metabolites of the central carbon metabolism were presented as the key intermediate of the cell metabolism, even if this treatment is limited to compounds presenting amino and/or carboxyl groups. We took a small aliquot from each sample to a mixed pool and the pooled sample was tested for matrix effects. After quality control (QC) without matrix effect, each sample was performed GC - MS analysis. More compounds and cleaner MS spectra were extracted using PARAllel FACtor analysis 2 (PARAFAC2) model [42] by MS-Omics software (www.msomics.com). Due to the derivatization, some extra peaks of compounds originated from the impurities in solvents, thus, these redundant peaks were removed. The GC-MS experiment was completed in MS-Omics company (Copenhagen, Denmark) using their standard system parameters.
41. Wikoff, W. R.; Anfora, A. T.; Liu, J.; Schultz, P. G. ; Lesley, S. A.; Peters, E. C.; Siuzdak, G. Metabolomics analysis reveals large effects of gut microflora on mammalian blood metabolites. Proc. Natl. Acad. Sci. 2009, 106, 3698-3703.
Reviewer 2 comment # 3:
· The authors also identified 13 unknown compounds, is anyways to make use of them?
Author response # 3: In this study, we only used the identified metabolites, so the 13 unknown compounds were discarded. Recently, some tools can use the peaks of unknown compounds for analysis, but the accuracy is a problem, so this study did not use them.
Reviewer 2 comment # 4:
· Did authors adjust for multiple testing when running the linear mixed model?
Author response # 4: This study tested breed, parity and RFI effects for 34 identified metabolites one by one, so the p-values should be close to q-values (FDR) for multiple testing correction. In this study, we used p-values to indicate the power of statistical analysis. Normally multiple testing correction is done in large scale omics experiments where several thousands to millions of statistical tests are conducted – here we have only 34 metabolites
Reviewer 2 comment # 5:
· Some metabolites have very low concentration and might costly to identify, how authors can use them as the biomarkers?
Author response # 5: Yes, this is a problem. However, these metabolites are not significant in the analysis; for example, malic acid is nearly zero in all the samples, so they will not be considered as the candidate biomarkers.
Reviewer 2 comment # 6:
· In the integration with the transcriptomic results, did the methods to generate results similar (for instance the model used and the fixed effects)?
Author response # 6: In the integration with transcriptomic results, the study focused on the integrated pathways and networks for biological processes. The fixed effects e.g., breed and parity were not used in integration analysis, but RFI (low and high) were used to find the upregulated/ downregulated gene-metabolite interaction network, as an upregulated and downregulated gene/metabolite was defined when logFC was positive and negative, respectively, after the low and high RFI comparisons. The results about this are shown in the Result part. Breed could also be analyzed with a larger sample size; otherwise, the dataset of this study will be too small to study all the interactions
Reviewer 2 comment # 7:
· The authors also need to pay an attention to the spelling and typo-errors.
Author response # 7: Thank you. We have improved English in the revised manuscript.
Reviewer 2 comment # 8:
· Did the authors check the hub (metabolites) or genes in the created network(s)?
Author response # 8: Yes. In the Figure 7, there is one upregulated gene (+HACL1) associated with two downregulated metabolites (-α-ketoglutarate and -Succinic acid).
3.3. Metabolic networks for gene expressions and metabolites
Two metabolites that were included in the metabolic network are expected to have an effect on feed efficiency (Fig. 7). 2-hydroxyacyl-CoA lyase 1 (HACL1), the first peroxisomal enzyme in mammals dependent on thiamin pyrophosphate (TPP), was negatively associated with α-ketoglutarate and Succinic acid in our results. α-ketoglutarate can determine the overall rate of TCA cycle, and modulate protein synthesis and bone development as an important source of amino acids for collagen synthesis in the cell and organism [33]. Succinic acid is also an intermediate compound of TCA cycle, which links and regulates energy metabolism like adenosine triphosphate (ATP) formation [34]. Through the mitochondrial Gamma-aminobutyric acid (GABA) transaminase, GABA forms glutamate and succinate semialdehyde with transamination of α-ketoglutarate [34], while GABA is converted from glutamate by the cytosolic glutamate decarboxylase [35]. The network of gene (HACL1) and metabolites (α-ketoglutarate and Succinic) involved in the key metabolic pathway of TCA cycle (Fig. 6) might be the potential biochemical mechanisms responsible for feed efficiency of dairy cows.
Reviewer 2 comment # 9:
Minors
Abstract
· Line 16: might replace “selection purposes” to “selection reducing feed intake” to make it clearly
· Line 17: this study is … should be this study was… to make it consistent to the whole manuscript
· Line 19-20: the author might specify more, how metabolites were analyses and which model used in very short, since it is the core section of the manuscript.
· Line 20: either use a linear regression model or linear regressions models
· Line 24-25: The first and the second principle component… can be written as “two major principle components”…
· Why no results reported for the linear mixed model here
Introduction
Line 42: should remove the recently, because the RFI was termed long ago, calculating can be replaced by subtracting or comparing
Line 47: “biological processes” should be replaced by “factors” since breed type… are not the processes
Line 49: Waghorn et al. (2006) [5] has reported that, remove has
Line 60: “Until now, metabolomics profiles associated with low and high RFI cows are scarce.” Could the authors explore more about it?
Line 61: this study is should written as this study was to make is consistent to the rest of manuscript
Results
The authors might provide the raw concentration results of the metabolite measure, since the figure make is quite difficult to examine the differences in concentrations among them
Line 88: which types of correlation
Line 97 and 102: be consistent in the tense used in the manuscript “Figure 3 provides” and “Figure 3 also revealed”
Line 111: From the figure 4, it is impossible to see the separation between two breeds
Figure 4: The authors should add the name on metabolites in the legend
Line 115-116: please paraphrase
Line 117: why 26
Table 2: please rename table, since it is confused that the RFI also significantly influenced on the metabolite concentrations
Line 146; check for double space
Line 154: should be was
Line 153-155: where we can find the result with HACL1
Discussion
161-174: Might move to introduction to give more space for deeper discussion
Line 178: Are differences between the metabolomics markers and the amino acid concentrations, since I found the authors mixed use of them in the manuscript?
Line 180: 34 or 37?
Could the authors comment about why fewer metabolites were identified in the study compared to the database?
Line 191: reviews to reviewed
Line 193: the prostate is mostly for the male reproduction, how it can link to the RFI in dairy cows or female cows here?
Can the authors have any comments why no metabolite significantly associated with RFI in the liner model test? Any potential biasness could affect the results of the test? How the relationship between the animals?
Line 196: there are many other metabolic studies, the authors might added some other to make the discussion deeper
Line 220: Many studies have linked the Gamma-aminobutyric acid (GABA) transaminase to the RFI as well as the feed intake, the authors might add more references and extend the discussion about them.
How the concentration of the two metabolites in plasma?
Methods
Could the authors explain why the cow number 3 are in the high RFI why the cow number 2 in the low RFI? When computing the RFI, did authors include other effects? It might be useful to give a very short summary of how the RFI was computed and explain in brief how they were was classified as low and high
When the plasma samples were collected? Do authors have any comments of about metabolite concentrations in different time of the day?
Line 268: “37 identified metabolites, 34 annotated metabolites and 13 unknown,” should it be 47
Line 287: were
Line 295: what is the metabolic value?
If the breeds are different in RFI, did the author try to run the model within each breed, since it might be useful to have different set of metabolites for each breed?
Why did not the authors include the unknown metabolites for the analyses, they might be useful in the future when more metabolites were identified?
Conclusions
The conclusion is too long, the authors already had the implication part, I don’t see the need for this part. Otherwise, please shorten it.
Author response # 9: Thanks a lot. We have corrected the manuscript as following these comments.
Why no results reported for the linear mixed model here
We have added the model results in the Abstract part.
Statistical results revealed a clearly significant differences between breeds, however, individual metabolites (Leucine, Ornithine, Pentadecanoic acid and Valine) association with RFI status were only marginally significant or not significant due to lower sample size.
Could the authors comment about why fewer metabolites were identified in the study compared to the database?
We have added the comment why fewer metabolites in the Discussion part.
Therefore, annotated metabolites were not used in this study. Afterwards, the identified metabolites tends to be a small size after rigorous analytical validation to exclude the annotated metabolites.
Line 193: the prostate is mostly for the male reproduction, how it can link to the RFI in dairy cows or female cows here?
Yes, it could be due to metabolites are enriched in mitochondria and peroxisome and then connected with prostate. Prostate, mitochondria and peroxisome metabolically interact with each other, and mitochondria and peroxisomes play major roles in cell metabolism as ubiquitous organelles, especially in terms of fatty acid metabolism.
Can the authors have any comments why no metabolite significantly associated with RFI in the liner model test? Any potential biasness could affect the results of the test? How the relationship between the animals?
In this study, we think that reason why no significant metabolite associated with RFI is mainly caused by the smaller sample size. The relationship between animals are from the similar genetic background and they are raised in the same experimental farm.
Line 196: there are many other metabolic studies, the authors might added some other to make the discussion deeper
Line 220: Many studies have linked the Gamma-aminobutyric acid (GABA) transaminase to the RFI as well as the feed intake, the authors might add more references and extend the discussion about them.
How the concentration of the two metabolites in plasma?
Thanks, we have added some other papers for deeper discussion. The concentration of the two metabolites is shown in Figure 7.
Furthermore, genome-wide association study (GWAS) revealed that RFI was in association with significant single nucleotide polymorphisms (SNPs) of GABRR2 gene, which can encode a receptor of GABA [36].
The two metabolites involved in the network showed higher values of α-ketoglutarate than Succinic acid (Fig. 7b).
Figure 7. (a) Gene-metabolite interaction network from combined metabolome and transcriptome studies. (b) The values of metabolites that are involved in the network.
Could the authors explain why the cow number 3 are in the high RFI why the cow number 2 in the low RFI? When computing the RFI, did authors include other effects? It might be useful to give a very short summary of how the RFI was computed and explain in brief how they were was classified as low and high
When the plasma samples were collected? Do authors have any comments of about metabolite concentrations in different time of the day?
Yes, we have added this information. Different metabolites should be different concentrations in the different time of the day, as the different metabolic processes varies in the different time.
The actual RFI value was defined by one-step approach using the method from Tempelman’s study [40] as reported by Salleh et al. (2017) [16]. Two cows including one low RFI cow and one high RFI cow were raised in one block, so the cow IDs were paired (Table 4). The cow was defined as low RFI when its actual RFI value was larger than the other cow in the same block, and then the paired cow was defined as high RFI accordingly.
Blood samples were collected to heparin tubes from the jugular vein of each cow at approximately 7 a.m.
If the breeds are different in RFI, did the author try to run the model within each breed, since it might be useful to have different set of metabolites for each breed?
Why did not the authors include the unknown metabolites for the analyses, they might be useful in the future when more metabolites were identified?
Yes, we have tried to separate the breed for analysis within each breed, but the results are not so good and as similar as the analysis together.
In addition, we have removed this sentence about unknown metabolites for clearer explanation.
The conclusion is too long, the authors already had the implication part, I don’t see the need for this part. Otherwise, please shorten it.
We have shortened the Conclusion part.

Reviewer 3 Report
This manuscript that has aimed to decipher metabolic biomarker for RFI for cattle management has been very poorly organized in terms of structure and language. Introduction part has been mostly straightforward although the last two paragraphs (line 55 to 76) needs significant re-writing. Authors should be more precise and clearer in their explanation of how metabolomics could be useful in their research goals, without being very elaborate. However, there are several other issues with this manuscript that need to be addressed before this can be considered for publication.
# Remaining part of the manuscript is not properly organized. Methods and material section need to be moved up so readers can actually follow where the significant metabolites shown in figures in the results have come from. Without this, following the result section is quite complicated. The ‘Method’ section also needs to present more information revealed, including how GCMS experiments were done. Only mentioning the derivatizing reagent is not enough and authors should describe how much sample of plasma was used, how much reagents were used, what are the parameters of GC and MS system used, column length etc.
#Authors have not mentioned how the protein in plasma was removed before GCMS and how it was ensured that metabolites were not lost through binding to residual protein removal.
# Table 4 should show the codes J1/H1 etc that are eventually being used in the result section. Even better way to label samples is using J1-High/ J1-Low/H1-High/H1-Low etc. This would make it much easier to corelate the tabulated data with the figures in result section.
# Most of the figures in the result section are not well discussed. This study looks more like a 2x2 study with two breeds and types of deit(high or low RFI). Thus, a two way ANNOVA would have been more appropriate to look at the variation of metabolite levels in these four groups and find significantly differentiating metabolites.
# It is unclear if the no of significant metabolites are 34 or 37. Both numbers have been used at different places in the manuscript. How these 34/37 metabolites were identified is not clear, weather those are from PCA loading score or from another test? Separation or clustering seen in PCA is not clearly convincing and providing results from PLSDA with permutation test could be more supportive. Authors have used Metaboanalyst for pathway analysis and correlation study, but they could have actually used it for PCA/PLSDA which would have provided box plots of differential levels of the significant metabolites among the four different sample groups.
# In Figure 6, what does the pathway impact value in x axis? It is not explained well. If the color of the circles denote the p values then what does the size of the circles mean here? These details are not well explained.
#Where is the figure and discussion about integration of the metabolomics data and transcriptomics data as mentioned in the abstract and introduction? Mixomics is another software that can give very nice correlation of multi omics data set and authors can explore it especially because the options for this in Metaboanalyst is quite limited.
# There are many grammatical error as well as improper sentence structure throughout the manuscript which must be corrected and re written.
Author Response
Reviewer 3
Reviewer 3 comment # 1:
This manuscript that has aimed to decipher metabolic biomarker for RFI for cattle management has been very poorly organized in terms of structure and language. Introduction part has been mostly straightforward although the last two paragraphs (line 55 to 76) needs significant re-writing. Authors should be more precise and clearer in their explanation of how metabolomics could be useful in their research goals, without being very elaborate. However, there are several other issues with this manuscript that need to be addressed before this can be considered for publication.
Author response # 1: Thank you for the comments. We have improved the structure and language in the revised manuscript. The last two paragraphs from line 55 to line 76 of Introduction part was also re-written.
Metabolomics has been increasingly used to measure the dynamic metabolic responses in dairy cows [13,14]. A metabolic pathway links series of chemical reactions in a cell. The pathways of metabolism enable to break down or synthesize many important molecules and initiate efficient reactions quickly. Metabolomics and pathways have been characterized for pregnant dairy cows to seek biochemical insight into possible biological modules related to early pregnancy [15]. Blood plasma metabolome can be measured on large number of animals cheaply compared to other -omics profiles used to predict RFI (e.g., RNAseq transcriptomics as in Salleh et al. (2017) [16]). Furthermore, metabolites are further downstream in the biological processes from DNA through RNA to proteins and hence closer to observable phenotypes. If metabolites are highly predictive of RFI phenotype, it could be used in animal selection of low RFI for better herd management or for breeding.
Until now, metabolic profiling of cows and availability of predictive metabolic biomarkers for RFI are scarce. Therefore, one of the objectives of this study was to generate a better knowledge of underlying metabolic mechanisms that possibly characterize low and high RFI Jerseys and Holsteins. The other objective was to identify potentially predictive metabolic biomarkers for RFIs. These objectives were achieved by designing cattle experiment to provide an evaluation and temporal comparison of the plasmatic metabolome analysis. By combining with pathway analysis methods, Gas chromatography - Mass spectrometry (GC-MS) system was used in the identification of each metabolite to generate a better understanding of the metabolic mechanisms occurring in Nordic dairy cows. Metabolite set enrichment analysis (MSEA) was used to investigate a set of functionally related metabolites. A novel two-way integration of metabolomics and transcriptomics profiles in low and high RFI cows was used to link their genomic variation with metabolomics variation and report upregulated and downregulated gene-metabolite combinations.
Reviewer 3 comment # 2:
# Remaining part of the manuscript is not properly organized. Methods and material section need to be moved up so readers can actually follow where the significant metabolites shown in figures in the results have come from. Without this, following the result section is quite complicated. The ‘Method’ section also needs to present more information revealed, including how GCMS experiments were done. Only mentioning the derivatizing reagent is not enough and authors should describe how much sample of plasma was used, how much reagents were used, what are the parameters of GC and MS system used, column length etc.
Author response # 2: The Materials and Methods section was in the fouth part as following the journal guidelines. Additionally, the “Method” section has been improved with more information.
In this study, Gas chromatography - Mass spectrometry (GC-MS) system was applied for identifying metabolites. GC-MS systems can only analyze volatile compounds, so chemical derivatization of nonvolatile compounds is required. With using methyl chloroformate (MCF) without any prior treatment, all plasma samples were derivatized to convert amino and nonamino organic acids into volatile carbamates and esters before GC - MS analysis. Accordingly, most metabolites of the central carbon metabolism were presented as the key intermediate of the cell metabolism, even if this treatment is limited to compounds presenting amino and/or carboxyl groups. We took a small aliquot from each sample to a mixed pool and the pooled sample was tested for matrix effects. After quality control (QC) without matrix effect, each sample was performed GC - MS analysis. More compounds and cleaner MS spectra were extracted using PARAllel FACtor analysis 2 (PARAFAC2) model [42] by MS-Omics software (www.msomics.com). Due to the derivatization, some extra peaks of compounds originated from the impurities in solvents, thus, these redundant peaks were removed. The GC-MS experiment was completed in MS-Omics company (Copenhagen, Denmark) using their standard system parameters.
In the GC-MS system, descriptive power (DP) was calculated as the ratio of standard deviation between experimental samples and QC samples. Generally, variables with a DP ratio higher than 2.5 are most likely to describe variation related to the experimental design. The percentage of precision (%) was also calculated to denote the relative standard deviation between QC samples, as a measure of the certainty of the individual compounds. The limit of detection (LOD) was listed to indicate the lowest value of a compound that the method enabled to detect.
Reviewer 3 comment # 3:
#Authors have not mentioned how the protein in plasma was removed before GCMS and how it was ensured that metabolites were not lost through binding to residual protein removal.
Author response # 3: The protein in plasma was treated by methyl chloroformate (MCF) without any prior treatment. Although this treatment is limited to compounds presenting amino and/or carboxyl groups, most metabolites of the central carbon metabolism were presented as the key intermediate of the cell metabolism.
With using methyl chloroformate (MCF) without any prior treatment, all plasma samples were derivatized to convert amino and nonamino organic acids into volatile carbamates and esters before GC - MS analysis. Accordingly, most metabolites of the central carbon metabolism were presented as the key intermediate of the cell metabolism, even if this treatment is limited to compounds presenting amino and/or carboxyl groups.
Reviewer 3 comment # 4:
# Table 4 should show the codes J1/H1 etc that are eventually being used in the result section. Even better way to label samples is using J1-High/ J1-Low/H1-High/H1-Low etc. This would make it much easier to corelate the tabulated data with the figures in result section.
Author response # 4: Thank you for the suggestions. We have replaced the codes J1/H1 by J1-High/ J1-Low/H1-High/H1-Low.
Table 4. Low/high residual feed intake (RFI) of Jersey and Holstein cows as described in Salleh et al., (2017).
Breed | RFI | actual RFI value | Cow ID | Parity | Breed | RFI | actual RFI value | Cow ID | Parity |
Jersey | Low | 0.80 | J1-Low | 1 | Holstein | Low | -0.03 | H2-Low | 1 |
2.23 | J3-Low | 3 | 0.10 | H4-Low | 2 | ||||
0.94 | J6-Low | 3 | 0.70 | H6-Low | 3 | ||||
0.46 | J8-Low | 2 | 0.89 | H7-Low | 2 | ||||
0.49 | J10-Low | 1 | 0.41 | H9-Low | 1 | ||||
High | -0.40 | J2-High | 1 | High | -1.10 | H1-High | 1 | ||
-0.04 | J4-High | 3 | 0.05 | H3-High | 3 | ||||
-1.05 | J5-High | 3 | -0.62 | H5-High | 3 | ||||
-1.71 | J7-High | 2 | -1.05 | H8-High | 2 | ||||
-0.51 | J9-High | 1 | -0.40 | H10-High | 1 |
Reviewer 3 comment # 5:
# Most of the figures in the result section are not well discussed. This study looks more like a 2x2 study with two breeds and types of deit(high or low RFI). Thus, a two way ANNOVA would have been more appropriate to look at the variation of metabolite levels in these four groups and find significantly differentiating metabolites.
Author response # 5: We have added more discussions about figures in the result section.
Thanks for this suggestion. We have tried to use two way ANOVA but have not found good results when we applied the RFI as low/high factors. Therefore, we used actual RFI values as regressions in the linear model.
Our results also revealed a good division of metabolites between Jersey and Holstein cows combining with low and high RFIs (Fig. 4). A total of 34 identified metabolites in four groups of low and high RFI Jersey and Holstein cows were clustered in three main categories, especially for the group of low RFI Holstein cows showing relatively higher values (Fig. 3).
where y is value of metabolite, is intercept, is Jersey and Holstein, is from 1 to 3, is actual RFI value as covariate and is a residual.
Reviewer 3 comment # 6:
# It is unclear if the no of significant metabolites are 34 or 37. Both numbers have been used at different places in the manuscript. How these 34/37 metabolites were identified is not clear, weather those are from PCA loading score or from another test? Separation or clustering seen in PCA is not clearly convincing and providing results from PLSDA with permutation test could be more supportive. Authors have used Metaboanalyst for pathway analysis and correlation study, but they could have actually used it for PCA/PLSDA which would have provided box plots of differential levels of the significant metabolites among the four different sample groups.
Author response # 6: Thanks. We have re-written the sentence about metabolite number and added the definition of identified and annotated metabolites.
Yes, MetaboAnalyst software can do PLS-DA and we have added the result of box plots using PLS-DA with four different groups (J-High/ J-Low/H-High/H-Low) in the manuscript.
Most of the values of 37 identified metabolites were higher than the values of LOD except 4-amino-benzoic acid, Cystine and Phosphoenolpyruvate that were all less than LOD, thus, this study only used 34 identified metabolites for the further analysis.
The identified metabolites were defined by comparing authentic chemical standards to the retention time and mass spectra. However, the annotated metabolites were primarily based on library matching of the acquired MS spectra with the National Institute of Standards and Technology (NIST) library.
Figure 3 provides clusters of metabolites with low and high RFI groups (all animals). In general, the concentrated values ranged from -1.5 to +1.5 after scaling by metabolite-wise in columns. Among 34 identified metabolites for which scaled values higher than LOD score, three main clusters were observed in the heat map. Lower cluster were all from fatty acids including Palmitoleic acid, Hexadecanoic acid, Octadecanoic acid, Heptadecanoic acid and Tetradecanoic acid (Fig. 3). The values of these metabolites were medium for low RFIs. High RFI Jersey and Holstein cows showed completely different values. Generally, metabolites of low RFI Holstein cows displayed higher values than the other three groups (high RFI Holstein, low RFI Jersey and high RFI Jersey) (Fig. 3)
Figure 3. Heat map for hierarchical clustering of 34 identified metabolites between low and high residual feed intakes (RFIs). Note: J/H with numbers indicates Jersey/Holstein ID. Low and High indicate low and high RFIs.
The first component (Component 1) and second component (Component 2) of PLS-DA analysis explained 61.5% and 11% variations of all 34 metabolites, respectively (Fig. 4). The high RFI group was shown in the horizontal line while low RFI group was in the vertical direction. It was observed that all of the metabolites were relatively more over represented in the Jersey group than the Holstein group. Additionally, a good division appeared between Jersey (J) and Holstein (H) breeds (Fig. 4).
Figure 4. Partial least squares - discriminant analysis (PLS-DA) of 34 identified metabolites between low and high residual feed intakes (RFIs). Note: J/H with number indicates Jersey/Holstein ID. Low and High indicate low and high RFIs.
Reviewer 3 comment # 7:
# In Figure 6, what does the pathway impact value in x axis? It is not explained well. If the color of the circles denote the p values then what does the size of the circles mean here? These details are not well explained.
Author response # 7: Yes. The color of the circles indicates the -log (P value) and the size of the circles indicates the matched metabolite ratio for each pathway. We have added this information in Figure 6.
Figure 6. Pathway analysis for 34 identified metabolites using Homo sapiens as library. (a) Single metabolic pathway analysis. (b) Integrated metabolic pathway analysis from combined metabolome and transcriptome studies. Note: The color and size of the circles indicate the -log (P value) and the matched metabolite ratio, respectively, for each pathway.
Reviewer 3 comment # 8:
#Where is the figure and discussion about integration of the metabolomics data and transcriptomics data as mentioned in the abstract and introduction? Mixomics is another software that can give very nice correlation of multi omics data set and authors can explore it especially because the options for this in Metaboanalyst is quite limited.
Author response # 8: We have added the Figure 7 about the integration of the metabolomics data and transcriptomics data. Yes, R package mixOmics can give the nice correlation multi-omics data by incorporating omics data of all the samples. However, this study would like to investigate the directions of genes-metabolite network, so we used the log of fold change (logFC) of RFI to define the direction as upregulated and downregulated gene/metabolite, when logFC was positive and negative, respectively, after the low and high RFI comparisons.
In addition, the gene-metabolite interaction network revealed only one network that was oneupregulated gene (+HACL1) associated with two downregulated metabolites (-α-ketoglutarate and -Succinic acid) (Fig. 7a). The two metabolites involved in the network showed higher values of α-ketoglutarate than Succinic acid (Fig. 7b).
Figure 7. (a) Gene-metabolite interaction network from combined metabolome and transcriptome studies. (b) The values of metabolites that are involved in the network.
3.3. Metabolic networks for gene expressions and metabolites
Two metabolites that were included in the metabolic network are expected to have an effect on feed efficiency (Fig. 7). 2-hydroxyacyl-CoA lyase 1 (HACL1), the first peroxisomal enzyme in mammals dependent on thiamin pyrophosphate (TPP), was negatively associated with α-ketoglutarate and Succinic acid in our results. α-ketoglutarate can determine the overall rate of TCA cycle, and modulate protein synthesis and bone development as an important source of amino acids for collagen synthesis in the cell and organism [33]. Succinic acid is also an intermediate compound of TCA cycle, which links and regulates energy metabolism like adenosine triphosphate (ATP) formation [34]. Through the mitochondrial Gamma-aminobutyric acid (GABA) transaminase, GABA forms glutamate and succinate semialdehyde with transamination of α-ketoglutarate [34], while GABA is converted from glutamate by the cytosolic glutamate decarboxylase [35]. Furthermore, genome-wide association study (GWAS) revealed that RFI was in association with significant single nucleotide polymorphisms (SNPs) of GABRR2 gene, which can encode a receptor of GABA [36]. The network of gene (HACL1) and metabolites (α-ketoglutarate and Succinic) involved in the key metabolic pathway of TCA cycle (Fig. 6) might be the potential biochemical mechanisms responsible for feed efficiency of dairy cows.
Reviewer 3 comment # 9:
# There are many grammatical error as well as improper sentence structure throughout the manuscript which must be corrected and re written.
Author response # 9: We have improved the written English. Please check the revised manuscript in the attachment.

Round 2
Reviewer 2 Report
The authors have been addressed my comments. Some minor suggestions
Line 22-23: Might re-write
Three main clusters were detected in the heatmap and all identified fatty acids (Palmitoleic, Hexadecanoic, Octadecanoic, Heptadecanoic and Tetradecanoic acid) were grouped in a cluster.
Change “more than 50%” to a majority (61.5%) of
Line 87: Don’t define the abbreviation in the heading
Line 97: The significant correlations of PCCs - Did authors mean the significance of the PCC, or did the authors compute significance of the correlation between different PCCs? Please clarify
Table1: Please add the note for star (*), does it mean significance?
Table 3: Define FDR in the note
Author Response
Reviewer 2
Reviewer 2 comment # 1:
The authors have been addressed my comments. Some minor suggestions
Line 22-23: Might re-write
Three main clusters were detected in the heatmap and all identified fatty acids (Palmitoleic, Hexadecanoic, Octadecanoic, Heptadecanoic and Tetradecanoic acid) were grouped in a cluster.
Change “more than 50%” to a majority (61.5%) of
Line 87: Don’t define the abbreviation in the heading
Line 97: The significant correlations of PCCs - Did authors mean the significance of the PCC, or did the authors compute significance of the correlation between different PCCs? Please clarify
Table1: Please add the note for star (*), does it mean significance?
Table 3: Define FDR in the note
Author response # 1: Thank you for the comments.
We have rewritten Line 22-23 and changed “more than 50%” to “a majority (61.5%) of”.
The abbreviation in the heading of Line 87 has been removed.
The significant correlations of PCCs mean the significance test of the PCC for each pair of input metabolites. We have rewritten this sentence in Line 97 and added the information in the Materials and methods part.
Yes, (*) means significance. We have added the note in the Table 1, Table 2 and Figure 2.
We have added the FDR definition in the Table 3.

Reviewer 3 Report
Authors have put significant efforts to address the concerns raised by all reviewers and this has sufficiently improved the clarity and quality of the manuscript. However, there are few points that still need to be considered and with those correction this manuscript is suitable for publication.
1) Figure 2 legend simply says it’s the PCC correlation of the 34 identified metabolites with values higher than LOD. However, neither the figure nor the following text mentions anything about what the *** starts in the figure means or what their number means? Also, it is not clearly communicated to readers, what does this PCC means when both axis have the same names of metabolites. This figure needs to be explained better.
2) Figure 4 is the 2D score plot of PLS-DA analysis, which surely shows good separation. However, unlike PCA, which is an unsupervised projection, PLS-DA is a supervised projection thus the model needs to be verified with a permutation test (available in Metaboanalyst) to confirm that the PLS-DA is valid with a p value<0.05 for permutation test. It would be good if the authors add this permutation test results to this figure.
Also, this figure is not the box plot as authors have mentioned in the response. A box plot here can be seen by going to the ‘loading plot’ of the PLS-DA analysis. Loading plot is a plot that demonstrates what are the metabolites that cause the separation between certain groups in PLS-DA. Clicking these dots shows a box plot demonstrating the relative fold change of that particular metabolite within those groups. Including couple of examples of box plots would be relevant but not absolutely necessary.
3) In the Figure 6, the meaning of ‘pathway impact value’ is still not clear, although the meaning of the color and sizes of the dots are clear. Table 3, lines 2 and 3 says metabolites match ratio for those 2nd and 3rd pathways are comparable 9/23 and 7/20. Yet their impact values are hugely different – 0.62 and 0.33. This needs some clarification.
Author Response
Dear Editors of Metabolites journal,
We have revised this manuscript according to the reviewers’ comments and responded them in this cover letter. All the comment-revised changes have been highlighted with the green color in the manuscript.
Sincerely,
The authors
Reviewer 3
Reviewer 3 comment # 1:
Authors have put significant efforts to address the concerns raised by all reviewers and this has sufficiently improved the clarity and quality of the manuscript. However, there are few points that still need to be considered and with those correction this manuscript is suitable for publication.
1) Figure 2 legend simply says it’s the PCC correlation of the 34 identified metabolites with values higher than LOD. However, neither the figure nor the following text mentions anything about what the *** starts in the figure means or what their number means? Also, it is not clearly communicated to readers, what does this PCC means when both axis have the same names of metabolites. This figure needs to be explained better.
Author response # 1: Thank you for the comments. We have added the explanation of *** and number in the figure note. The PCCs of diagonal when both axis have the same metabolites names were 1.
Figure 2. Pearson correlation coefficient (PCC) analysis of 34 identified metabolites. Note: *indicates P value < 0.05, **indicates P value < 0.01 and ***indicates P value < 0.001. The number on right bar indicates the PCCs from -1 to 1. The PCCs of the diagonal were 1.
Reviewer 3 comment # 2:
2) Figure 4 is the 2D score plot of PLS-DA analysis, which surely shows good separation. However, unlike PCA, which is an unsupervised projection, PLS-DA is a supervised projection thus the model needs to be verified with a permutation test (available in Metaboanalyst) to confirm that the PLS-DA is valid with a p value<0.05 for permutation test. It would be good if the authors add this permutation test results to this figure.
Also, this figure is not the box plot as authors have mentioned in the response. A box plot here can be seen by going to the ‘loading plot’ of the PLS-DA analysis. Loading plot is a plot that demonstrates what are the metabolites that cause the separation between certain groups in PLS-DA. Clicking these dots shows a box plot demonstrating the relative fold change of that particular metabolite within those groups. Including couple of examples of box plots would be relevant but not absolutely necessary.
Author response # 2: Thank you. We have added the permutation test results to Figure 4. We also included the loading plot and two examples of box plots in the Figure 4.
The loading plot results showed that eight metabolites (Citric acid, Heptadecanoic acid, Hexadecanoic acid, Octadecanoic acid, Palmitoleic acid, Pentadecanoic acid, Tetradecanoic acid and Valine) caused the separation between different breed and RFI groups in PLS-DA (Fig. 4b). As a supervised method, PLS-DA is more susceptible to overfitting, so it needs to be verified. The permutation results here confirmed that the PLS-DA was valid with a P value (0.012) < 0.05 after 1,000 permutation tests (Fig. 4c). From the box plots of α-ketoglutarate and Succinic, the fold change of low RFIs showed relatively higher values than the fold change of high RFIs (Fig. 4d and Fig. 4e).
Figure 4. Partial least squares - discriminant analysis (PLS-DA) of 34 identified metabolites between low and high residual feed intakes (RFIs) of Jersey and Holstein cows. (a) 2D-score plot of PLS-DA. (b) Loading plot of metabolite separations in PLS-DA. (c) Permutation test (n = 1,000) for PLS-DA. (d) and (e) Box plots of α-ketoglutarate and Succinic acid for relative fold change after PLS-DA. Note: J/H with number indicates Jersey/Holstein ID. Low and High indicate low and high RFIs.
Reviewer 3 comment # 3:
3) In the Figure 6, the meaning of ‘pathway impact value’ is still not clear, although the meaning of the color and sizes of the dots are clear. Table 3, lines 2 and 3 says metabolites match ratio for those 2nd and 3rd pathways are comparable 9/23 and 7/20. Yet their impact values are hugely different – 0.62 and 0.33. This needs some clarification.
Author response # 3: The definition of pathway impact value has been added in the Figure 6. As the calculation of match ratio and pathway impact value are different, the line 2 and line 3 of Table 3 showed different impact values.
Figure 6. Pathway analysis for 34 identified metabolites using Homo sapiens as library. (a) Single metabolic pathway analysis. (b) Integrated metabolic pathway analysis from combined metabolome and transcriptome studies. Note: The color and size of the circles indicate the -log (P value) and the matched metabolite ratio, respectively, for each pathway. Pathway impact value is calculated as the sum of importance measures of the matched metabolites divided by the sum of the importance measures of all metabolites.

Round 3
Reviewer 2 Report
Thank you. All my comments have been addressed.
Reviewer 3 Report
Thank you for addressing all the concerns. I would recommend accepting for publication in this current form.